

# Wet and dry seasons modulate coastal coccolithophore dynamics off South-western Nigeria (Gulf of Guinea)

Falilu Adekunbi[1,2,3], Michaël Grelaud[3], Gerald Langer[3], Lucian Chukwu[1], Marta Alvarez[4], Shakirudeen Odunuga[5], Kai G. Schulz[6], Patrizia Ziveri[3,7,8]

[1]Department of Marine Sciences, Faculty of Science, University of Lagos (UNILAG), Lagos, Nigeria
[2]Nigerian Institute for Oceanography and Marine Research, (NIOMR) Lagos, Nigeria
[3]Institut de Ciència i Tecnologia Ambientals (ICTA-UAB), Universitat Autònoma de Barcelona, Barcelona, Spain
[4]Instituto Español de Oceanografía (IEO-CSIC), A Coruña, Spain
[5]Department of Geography, Faculty of Social Sciences, University of Lagos (UNILAG), Lagos, Nigeria
[6]Centre for Coastal Biogeochemistry, School of Environment Science and Engineering, Southern Cross University, Lismore, NSW 2480, Australia
[7]Catalan Institution for Research and Advanced Studies (ICREA), Barcelona, Spain
[8]BABVE Dept., Universitat Autònoma de Barcelona, Barcelona, (ICTA-UAB), Spain

*Correspondence to*: Michaël Grelaud (michael.grelaud@uab.cat)

**Abstract.** Coccolithophores are calcifying unicellular phytoplankton at the base of the marine food web, playing a key role in pelagic calcium carbonate production. While their sensitivity to environmental change is well established, their ecological importance in tropical coastal systems remains underexplored, particularly along the African coastline. Here, we present the first multi-seasonal assessment of living coccolithophore communities off Lagos, southwest Nigeria, in the Gulf of Guinea.

Periodic sampling was conducted at three coastal stations from December 2018 to April 2021 to evaluate species composition, standing stocks, diversity, and ecological drivers. Coccolithophore abundances showed clear seasonal patterns, with significantly higher ($p < 0.05$) standing stocks and diversity during the wet season. Total abundances ranged from $0.3 \times 10^3$ cells $L^{-1}$ in the dry season to $5.5 \times 10^3$ cells $L^{-1}$ in the wet season, with *Gephyrocapsa oceanica* dominating dry periods and *Emiliania huxleyi* prevailing during the wet season. Seasonal changes were linked to the migration of the Inter-tropical

Convergence Zone (ITCZ), which modulates precipitation and current direction along the Gulf of Guinea. Interestingly, chlorophyll-a concentrations appeared decoupled from coccolithophore abundance, suggesting other phytoplankton groups may drive primary productivity in this region. Despite regional differences in oceanographic settings, the observed standing stocks fall within the global range of coastal coccolithophore assemblages, supporting the hypothesis that these communities are shaped by a set of common ecological constraints. As tropical coastal regions al-ready face multiple pressures from climate

change, projected southward shifts of the ITCZ could alter precipitation regimes and current dynamics, with potential implications for coccolithophore community composition and coastal biogeochemical cycling.



## 1 Introduction

Coccolithophores are a dominant group of calcareous phytoplankton. They play an important role in global biogeochemical
cycling and account for up to 90% of pelagic calcium carbonate production in the open ocean (Ziveri et al. 2023). They are a
key component of the marine food web, and their significance in open ocean areas has long been recognised (Balch et al.,
2019; Li et al., 2024; Poulton et al., 2017). More recently their ecological role in coastal areas has been emphasized (e.g.,
Addante et al., 2023; Godrijan et al., 2018), as coastal waters often display considerable seasonal and inter-annual variability.
This is due to the interplay of land-sea physical and biogeochemical processes such as water mass mixing, coastal
geomorphology, river inputs and runoff, as well as variability in primary productivity and respiration (Carstensen et al., 2019),
coastal environmental parameters that mainly drive coccolithophore seasonality vary with geographic and oceanographic
settings (Godrijan et al., 2018; Keuter et al., 2023; Priyadarshani et al., 2019). Sometimes dominating effects can be identified,
e.g. seasonal upwelling (Ausín et al., 2018; Fiúza, 1984; Moita et al., 2003; Silva et al., 2008), or seasonal mixing (Keuter et
al., 2022; Šupraha et al., 2016). However, despite the crucial role  coccolithophores play as marine primary producers, a limited
number of studies have addressed their region-specific seasonal variability in neritic environments (Bonomo et al., 2014,
2018a, 2018b; O'Brien et al., 2013). This is especially true for African coastal areas, which are often overlooked in
coccolithophores research. In particular, no study to date has investigated coccolithophores in the Gulf of Guinea, along the
western coast of central Africa, where previous research has instead focused on diatoms and dinoflagellates as the dominant
primary producers (Anang, 1979; Issifou et al., 2014; Koffi et al., 2024; Seu-Anoï et al., 2011).
Climate change and anthropogenic activities are profoundly altering coastal ecosystems (He et al., 2019), especially in tropical
regions (Kleypas, 2019), where hydrographic conditions are closely linked to atmospheric circulation patterns (Li et al., 2016).
In the Gulf of Guinea, climate is strongly influenced by the Intertropical Convergence Zone (ITCZ), a region where the
Northeast and South-east trade winds converge. The region exhibits pronounced seasonality, primarily driven by ITCZ shifts
caused by differential heating of the land and ocean, which strongly influence rainfall patterns (Kang, 2020; Nwankwo, 1996;
Odekunle et al., 2008). This mechanism controls the location of the sub-tropical Atlantic biogeochemical divide, with a high-
phosphate, low-iron system in the south, and a low-phosphate, high-iron system in the north (Schlosser et al., 2014). Recent
projections suggest that the ITCZ is expected to narrow and shift southwards due to ongoing climate change (Mamalakis et
al., 2021) while local observations have demonstrated a shift in rainfall pattern, intensity and frequency (Fasona et al., 2019),
with potentially significant implications for tropical climate regimes and marine productivity. In the nearby Cabo Verde region,
northwest of the Gulf of Guinea, previous studies have shown that the seasonal migration of the ITCZ is a key driver of
coccolithophore species distribution (Narciso et al., 2021; Silva et al., 2013). This underscores the urgent need to better
understand the dynamics and response of the primary producers, which form the base of the food web, to such environmental
changes. In this study, for the first time the coccolithophore community and associated environmental parameters were
periodically monitored in the coastal area off Lagos, Nigeria, from December 2018 to April 2021, covering both the wet and
dry seasons, characteristic of the area.





## 2 Materials and methods

### 2.1 Study area

The study area is situated off the coast of Lagos (southwestern Nigeria) in the Gulf of Guinea (Fig. 1). The main oceanographic settings in the area are modulated by the equatorial currents of the East-ern Equatorial Atlantic Ocean: the equatorward Canary

Current (CC) flowing along the western coasts of North Africa connects to the eastward North-Eastern Counter Current (NECC). The east-ward Guinea Current (GC), which is an extension of the NECC (Hisard et al., 1980), flows along the coast of West Africa at a latitude of 4.5°N (Fig 1). The coastal GC is dominant through the year, reaching a maximum velocity of 1 m.s-1 from August to September in the vicinity of Cape Palmas in Côte d'Ivoire up to the West of Lagos (Da-Allada et al., 2021; Djakouré et al., 2017; Sohou et al., 2020). The GC transports warm (23-32 °C) and less salty water masses whose

variability is driven by precipitation, wind stress and diffusion due to vertical shear (Alory et al., 2021; Awo et al., 2018; Binet, 1997). Off the Cameroun coast, the steep continental slope causes a reflection of the GC towards the west, forming the South Equatorial Current (SEC). Close to the SEC is a sub-surface eastward flow of the Equatorial Under Current (EUC) (van Bentum, 2012). During major upwelling in boreal summer (July to September) shoaling of the thermocline, brought about by the strengthening of the GC, drives a cooling of the surface ocean along the coast of Côte d'Ivoire and Ghana with a weak

signal on the Western flank of Lagos coast (Oyewo et al., 1982; Sohou et al., 2020). This seasonal coastal cooling influences regional climate and also transports carbon-rich deep waters that seasonally drive variability in carbonate chemistry (Koffi et al., 2016). Primary production, also fuelled by upwelling, supports fisheries in the Gulf of Guinea Djakouré et al., 2014; Zeller et al., 2020).

The rainfall in the Lagos coastal environment is characterized by a double maximum pattern where according to Nwankwo

(1996) the dry season commences in November and extends until April while the wet season is from May to October with a break in August (Fig. 1). However, climate change has resulted already in local shifts in rainfall pattern, intensity and frequency, consequently the wet sea-son now sets in from March to November reaching a peak in July and September (Fasona et al., 2019). Seasonality in rainfall in this rainforest zone is brought about by continental scale movement of the ITCZ due to shifts in tropical maritime and continental air masses (Odekunle et al., 2008) as well as seasonal migration of the south-east

trade wind system (Nwankwo et al., 2003).



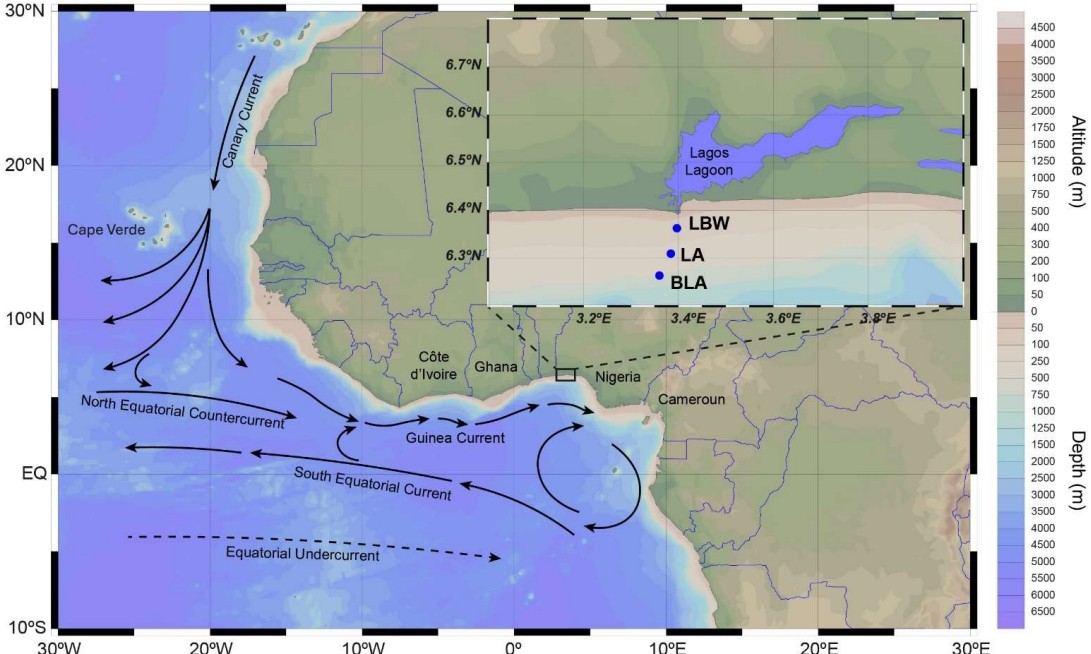

Figure 1: General and local study area maps. The main surface currents in the Gulf of Guinea are presented in the general map (modified from Diop et al., 2014), while the 3 study sites are presented on the zoomed map in the upper right corner: Lagos Break-water (LBW), Lagos Anchorage (LA) and Beyond Lagos Anchorage (BLA). Map produced with Ocean Data View (Schlitzer 2023).

## 2.2 Sampling strategy

Discrete samples of seawater were collected at three stations off Lagos in the Gulf of Guinea, onboard the M/V Sea Marshal of the Nigerian Maritime Administration and Safety Agency (NI-MASA) (Fig.1, Table 1). The three stations, Beyond Lagos Anchorage (BLA), Lagos Anchorage (LA) and Lagos Break-waters (LBW) are respectively situated at 33.4, 22.2 and 11.1 km offshore, south of Lagos Harbour (Fig. 1). A total of 13 sampling campaigns were carried out between December 2018 and April 2021. A total of 99 samples were collected across the three stations at consistent depths ranging between 5 and 50 m (Table 1). The water samples were collected with a 2.5L Teflon coated Niskin bottle (Model 1010, General Oceanics, Miami, FL, USA) with a graduated cable fastened to a 3kg weight to allow sinking to target depth. A multiparametric YSI model Pro 1030 probe with an extended cable was used to measure the temperature (±0.2ºC) and the salinity (±0.1ppt) at each target depth. Discrete seawater samples at target depths were taken to assess the coccolithophore community and target environmental variables (dissolved inorganic nutrients, chlorophyll-a concentrations and seawater carbonate chemistry).



Table 1: Station names, coordinates, sea floor depth and target sampling depths.

| Station Name | Latitude | Longitude | Bottom Depth (m) | Sampling Depths (m) |
|---|---|---|---|---|
| Beyond Lagos Anchorage (BLA) | 6.26278 | 3.36037 | 51 | 5/25/50 |
| Lagos Anchorage (LA) | 6.30861 | 3.38417 | 36 | 5/25/35 |
| Lagos Break-water (LBW) | 6.36222 | 3.39750 | 17.6 | 5/15 |

**2.3 Coccolithophore community**

The seawater samples for coccolithophore community composition were stored shortly (<5 hours) in pre-cleaned containers before filtering them in the laboratory. For each sample, an average of 2 litres of seawater was filtered on a polycarbonate membrane (0.4μm pore size, 47mm diameter) for Scanning Electron Microscope (SEM) analyses, and cellulose acetate membrane (0.45μm pore size, 47mm diameter) for polarized light microscopy. The samples were then oven-dried at 60°C for one hour before storage in plastic Petri dishes.

Coccolithophore quantification for standing stock and taxonomy was performed on an 8mm × 8mm piece of filter mounted on glass slides with Canada Balsam. The counting was performed with a ×1000 magnification Olympus polarized light microscope (model BX-51 U-FMP). A minimum of 100 (up to 513) coccospheres were counted per sample. The taxonomic identification was carried out following the coccolithophore light microscopy guide by Frada et al. (2010), the Nannotax3 website (http://ina.tmsoc.org/Nannotax3/) and further confirmed with SEM. Based on coccosphere counts, absolute density in

cells/l was calculated following the equation (Oviedo et al., 2014; Ziveri et al., 1995):

$$CD = \left(\frac{A \times N}{a}\right) \div V$$

Where: CD = Coccospheres concentration (cell/l)

A = Effective filtered area (mm2)

N = Number of coccospheres counted

a  = Analyzed area (mm2)

V = volume of water filtered (l) young

For SEM, a random section of the filter was stuck on stubs and coated (EMITECH K550X) with 95% Gold and 5% Palladium at a current of 20 mA for 3 minutes. Taxonomic identifications were performed with a MERLIN SEM at a magnification of ×3000, following the identification guide by Young et al. (2003).2.4 Carbonate chemistry variables

**2.4 Carbonate chemistry variables**

Seawater samples for carbonate chemistry analysis were collected in acid-washed borosilicate bottles and preserved with HgCl₂ following Dickson et al. (2007). Analyses were conducted at the INOCEN laboratory (IEO-CSIC, Spain) within six months of collection. For 2018–2020 samples, TA and DIC were measured; while for 2021, TA and pH were analysed due to equipment issues. DIC was measured coulometrically with a VINDTA 3D system, calibrated using certified reference material (CRM;



batch #177), with an accuracy of ±2.0 µmol kg⁻¹. TA was determined via potentiometric titration using a Titrando 909 system and CRM (batch #190), achieving ±2 µmol kg⁻¹ precision. pH was measured spectrophotometrically with m-cresol purple and a SHIMADZU UV-2600, at $25 \pm 0.2°C$, yielding ±0.002 pH unit accuracy. All pH values are reported on the total scale for in situ conditions. See Supplementary Material for more details.

### 2.5 Dissolved inorganic nutrients

Dissolved inorganic nutrients samples were subsampled from the polyethylene bottles collected for the coccolithophore community sampling and an aliquot was stored at 4°C until analyses in less than 24 hours. The samples were filtered through a 0.45µm Glass Fibre filter and the filtrate was used for the determination of $NO_2^-$, $NO_3^-$, $PO_4^{3-}$, DIN ($NO_3$-N, $NO_2$-N, and $NH_3$-N), and silicate ($Si(OH)_4$) using a HACH DR 3900 spectrophotometer following (Rice et al., 2012).

### 2.6 Chlorophyll-*a*

For each sample, 200mL of seawater was filtered through a Whatman GF/F, glass fibre filter (nominal pore size 0.7 µm, 47 mm diameter). If immediate filtration was not possible, the water samples were kept on ice in the dark and filtered within 24 hours. Filter homogenization and pigment extraction was carried out with 90% acetone overnight in the dark at 4 °C, following standard protocols (Arar et al., 1997). The concentration of Chlorophyll-*a* (Chl-*a*) was then determined with a Lamotte Smart Spectro spectrophotometer. Results were validated with calibration standards and squared correlation coefficients $r^2 > 0.9990$
were considered acceptable (Panagiotopoulos et al., 2009).

### 2.7 External datasets integration

To put our results into perspective, existing dataset were used and processed. Monthly precipitation (mm/hr) between 20ºN and 10ºS over the study area, was retrieved from the Goddard Earth Sciences Data Information Services Center (GES DISC, Huffman et al., 2019). The data collected covers the study period with a resolution of 0.1º × 0.1º. The data of the monthly
eastward seawater velocity were retrieved from the Copernicus Marine Environment Monitoring Service (CMEMS) through the Global Ocean Ensemble Physics Reanalysis (2023). They cover the study period with a resolution of 0.25º × 0.25º. Both datasets were extracted with SeaDAS (version 8.4.0, https://seadas.gsfc.nasa.gov/) using the pixel extraction tool. For precipitation, the data for each pixel in the area of interest was extracted for each month, while for the eastward sea water velocity, the data at the location of each station was extracted within a window size of 5 × 5 pixels (i.e.: 1.25º × 1.25º). The
data extracted were processed and mapped with Ocean Data View (Schlitzer, 2023).
Global datasets for coccolithophore standing stocks (de Vries et al., 2020; O'Brien, 2012) were retrieved and process in order to compare with our results. Only samples collected in coastal areas (<50km from the coastline) and with a maximum seafloor depth of 100m, to match the characteristics of our study sites (Table 1), were processed. The selected samples were then organized and averaged for major oceanic regions (see supplementary material for more details).2.8 Statistical analyses



The study area is situated off the coast of Lagos (southwestern Nigeria) in the Gulf of Guinea (Fig. 1). The main oceanographic settings in the area are modulated

**2.8 Statistical analyses**

To test whether environmental parameters and coccolithophore community composition differed between the dry and wet seasons, and to identify which environmental conditions may influence coccolithophore communities, statistical analyses were

performed using RStudio (v4.3.1; RStudio Team, 2020). The Shapiro-Wilk test was used to assess the normality of each variable within each season (Table S2). When normality was observed for a given parameter or species in both seasons, Welch two-sample t-test was applied to compare means between seasons (Table S3). When normality was not met for at least one season, the non-parametric Wilcoxon rank-sum test was used instead (Table S4). Finally, a principal component analysis (PCA) and Pearson correlations were performed between the environmental parameters and the coccolithophore data to explore

patterns and interactions. See Supplementary Material for more details.

**3 Results**

**3.1 Coccolithophore distribution**

Only heterococcolithophores were identified in all the samples (15 taxa, Table S1), as no holococcolithophore could be observed. Total coccosphere abundances varied through the studied period from a minimum of $0.3 \times 10^3$ cells L$^{-1}$ which

coincided with the dry season to a maximum of $47.5 \times 10^3$ cells L$^{-1}$ in the wet season. Wet season abundances were on average higher ($17.3 \times 10^3 \pm 11.4 \times 10^3$ cells L$^{-1}$) compared to dry season records ($10.6 \times 10^3 \pm 9.2 \times 10^3$ cells L$^{-1}$) (Fig. 2d and f). In general, the average abundances were higher at the mid-point station LA ($14.4 \times 10^3 \pm 12.3 \times 10^3$ cells L$^{-1}$), followed by the farthest station BLA ($13.9 \times 10^3 \pm 10.0 \times 10^3$ cells L$^{-1}$) and the coastal station LBW ($11.3 \times 10^3 \pm 9.31 \times 10^3$ cells L$^{-1}$) (Fig. 2b).

Among the 15 identified taxa, only 7 exceeded 1% relative abundance (Table S5). The assemblages were dominated by *Emiliania huxleyi* and *Gephyrocapsa oceanica*, together accounting for 83.2% of the community (Fig. 2a). *Emiliania huxleyi* dominated during the wet season, representing an average of 66.5% of the assemblage with a mean concentration of $11.4 \times 10^3 \pm 9.0 \times 10^3$ cells L$^{-1}$ (range: $1.9 - 42.6 \times 10^3$ cells L$^{-1}$) (Fig. 2e and f). During the dry season, *G. oceanica* dominated, representing an average of 63.7% of the assemblage with a mean concentration of $6.7 \times 10^3 \pm 7.5 \times 10^3$ cells L$^{-1}$ (range: $0.1 -$

$38.0 \times 10^3$ cells L$^{-1}$) (Fig. 2c and d). During the dry season, only 3 other species exceeded 1% of the assemblages: *Gephyrocapsa ericsonii* (11.3%, $3.0 \times 10^3 \pm 4.2 \times 10^3$ cells L$^{-1}$), *Calciopapus rigidus* (1.7%, $0.3 \times 10^3 \pm 0.3 \times 10^3$ cells L$^{-1}$) and *Syracosphaera tumularis* (1.2%, $0.26 \times 10^3 \pm 0.23 \times 10^3$ cells L$^{-1}$) (Fig. 2c and d). During the wet season, 4 other species exceeded 1% of the assemblage: *S. tumularis* (8.2%, $2.93 \times 10^3 \pm 5.93 \times 10^3$ cells L$^{-1}$), *Umbellicosphaera hulburtiana* (3.0%, $0.67 \times 10^3 \pm 0.85 \times 10^3$ cells L$^{-1}$), *C. rigidus* (2.8%, $0.73 \times 10^3 \pm 0.64 \times 10^3$ cells L$^{-1}$) and *Discosphaera tubifera* (1.2%, 0.59

$\times 10^3 \pm 1.10 \times 10^3$ cells L$^{-1}$) (Fig. 2e and f).



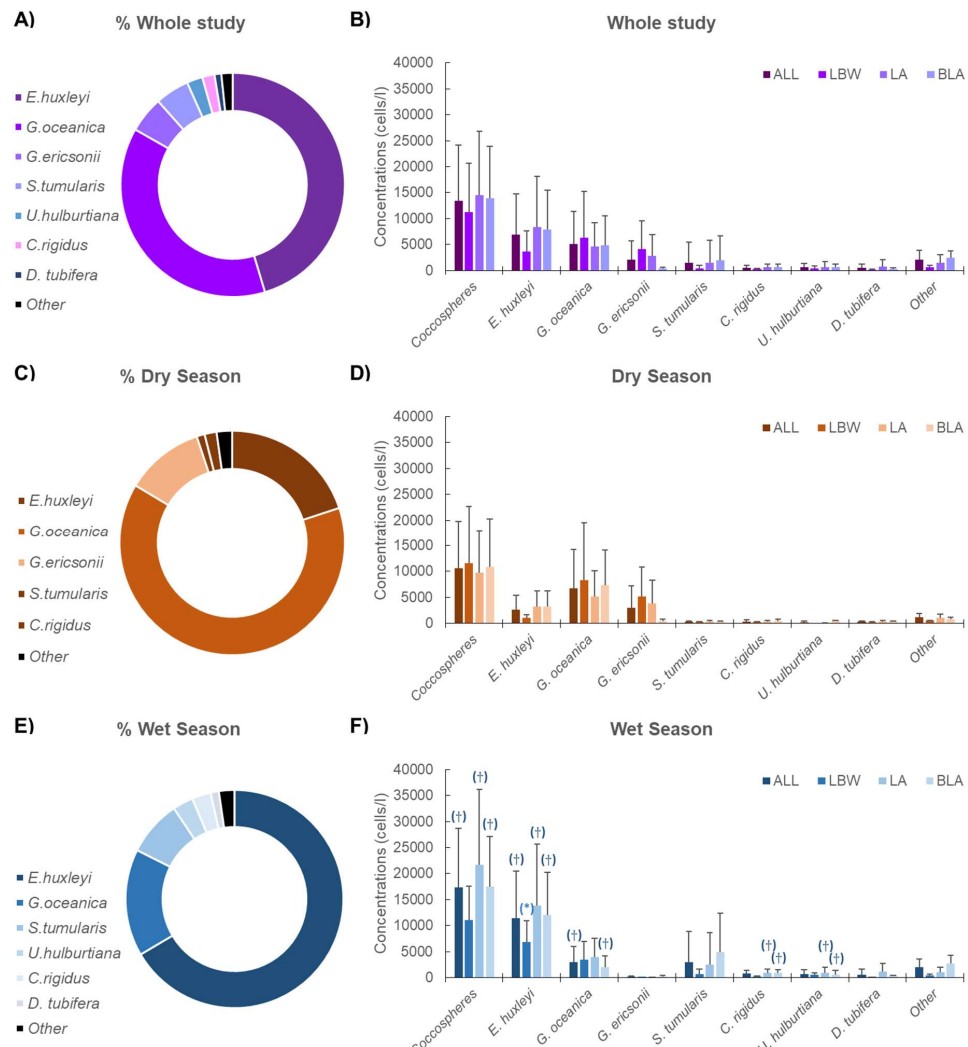

**Figure 2: Coccolithophore distribution during the dry and wet seasons:** Left panels (A, C, E), from top to bottom, show the average relative abundances of the dominant species (contributing >1% of the coccolithophore assemblage) for (A) the whole study period, (C) the dry season and (E) the wet season respectively. Right panels, from top to bottom, show the average absolute abundances of the dominant species for the 3 stations together (ALL) and each individual station (LBW, LA and BLA), for (B) the whole study period, (D) the dry season and (F) the wet season respectively. The error bars (1σ standard deviation) only show the upper bound for visibility. Significative difference (p-value <0.05) between the two seasons is given by the (*) and the (†) on the lower right panel, respectively for the Welch two-sample test and the Wilcoxon sum rank test (see section 2.8 Statistical analysis for more details).




Spatially, *E. huxleyi* reached its peak abundance at the intermediate LA station during the wet season (Fig. 2f), while *G.*
*oceanica* was most abundant at coastal station LBW during the dry season. For the minor species, *G. ericsonii* is the only one
to show its highest abundances during the dry season at the coastal LBW station. The other minor species have all their highest
abundances during the wet season: *S. tumularis* and *C. rigidus* at the offshore station BLA, and *D. tubifera* and *U. hulburtiana*
at the intermediate station LA. This is reflected in a higher species richness during the wet season for each station (Table 2),
as well as in coccolithophore diversity with higher Shannon diversity index (H') during the wet (H'=1.62) compared to the dry
season (H'=1.23) (Tables 2). The highest diversity for the wet season was recorded at the intermediate station LA (H'=1.76)
and at the most coastal LBW station for the dry season (H'=1.28). Although the coccolithophore diversity in the coastal waters
of the Gulf of Guinea are relatively low, it remains in the same range as other coastal sites (e.g. Balestra et al., 2008, H'= 0.00
– 2.17; Dimiza et al., 2014, H'= 0.14 – 1.42; Luan et al., 2016, H' = 0.7 –3.3).

**Table 2: Mean Shannon index (SI) and samples richness (Rich.) for the 3 stations and all together during the dry and wet season.**
**For richness, the values into parenthesis represent the highest richness recorded. Station labels in Table 1.**

| | LBW | | LA | | BLA | | All | |
|---|---|---|---|---|---|---|---|---|
| | SI | Rich. (max) | SI | Rich. (max) | SI | Rich. (max) | SI | Rich. (max) |
| Dry season | 1.28 | 4.2 (7) | 1.12 | 4.1 (8) | 1.20 | 4.5 (8) | 1.23 | 4.2 (9) |
| Wet season | 1.43 | 4.8 (9) | 1.76 | 6.2 (9) | 1.63 | 5.7 (10) | 1.62 | 5.6 (10) |

### 3.2 Environmental variables

In general, the environmental variables measured (Table S6) along with the coccolithophore concentrations showed minimal
seasonal variations with statistically significant differences mainly for salinity, silicate and precipitation (Fig. 3, Table S3,
Table S4). Slight differences, though not significant, were also observed: the temperature and the nitrate concentrations were
the only variables that had slightly higher average values during the dry season, with 28.25 ± 1.70 °C and 32.34 ± 24.75
µmol/kg respectively, compared to the wet season, with 27.80 ± 2.49 °C and 26.19 ± 19.20 µmol/kg respectively (Fig. 3a and
h). In contrast DIC, pH and nitrite concentrations showed an opposite trend with slightly higher values during the wet season,
with 2013 ± 112 µmol/kg, 7.94 ± 0.1 and 0.88 ± 1.80 µmol/kg respectively, compared to the dry season with 33.01 ± 2.03 psu,
2004 ± 112 µmol/kg, 7.93 ± 0.1 and 0.40 ± 0.73 µmol/kg respectively (Fig. 3b, d, f and i). The chlorophyll-*a* and phosphate
concentrations had on average very similar values between the dry and wet season (Fig. 3c and g), with respectively 11.34 ±
6.18 µg/l and 11.35 ± 5.72 µg/l for the former and 2.28 ± 3.78 µmol/kg and 2.25 ± 3.23 µmol/kg for the latter. There were
differences between the 3 stations as most of the time, they showed spatial and temporal variations with no consistent pattern.
As an ex-ample, temperature (Fig. 3a), pH (Fig. 3f), and chlorophyll-a (Fig. 3c) were on average higher during the dry season
at the coastal LBW and intermediate LA stations, while for the offshore BLA station, the waters are warmer during the wet
season (Fig. 3a).



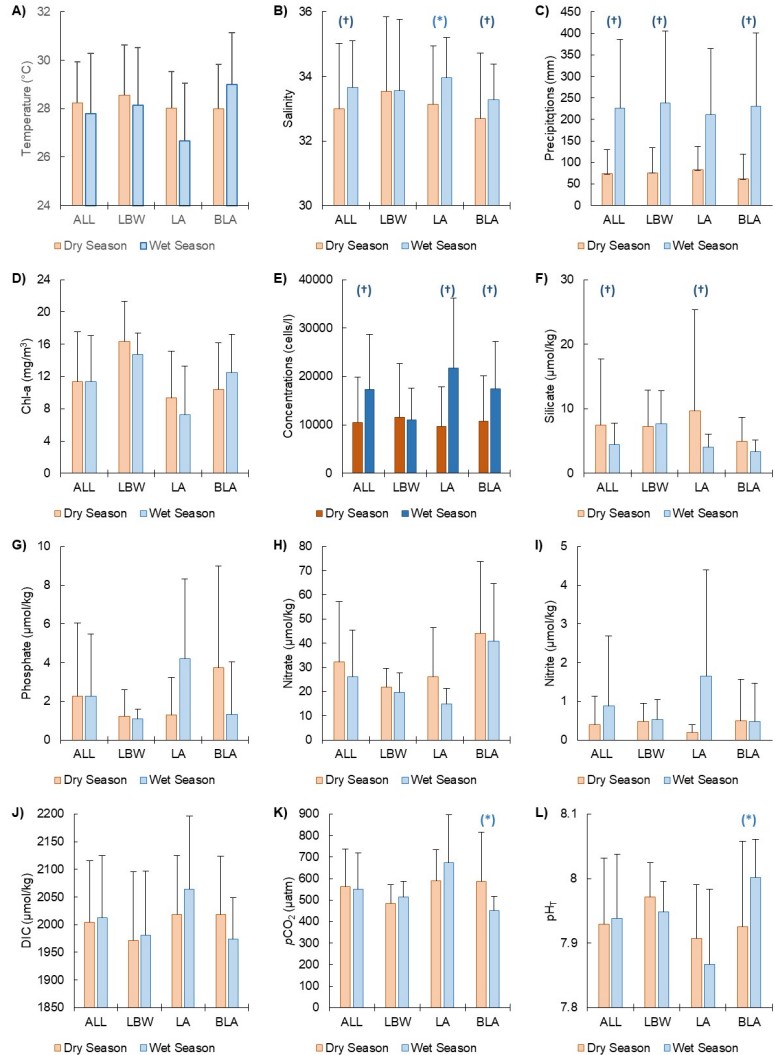

**Figure 3: An Environmental parameter during the dry and wet seasons: Mean dry and wet season environmental parameters for the three sampled stations together (ALL) and each individual station (LBW, LA and BLA see Table 1). (A) Temperature, (B) Salinity, (C) Precipitations, (D) Chlorophyll a (Chl-a), (E) Coccosphere concentrations, (F) Silicate, (G) Phosphate, (H) Nitrate and (I) Nitrite, (J) Cissolved inorganic carbon (DIC), (K) Partial pressure of carbon dioxide (pCO2) and (l) pH on the total scale at 25ºC (pHT). The central panel (coccosphere concentrations) have darker colours to facilitate the comparison with the environmental parameters. The error bars (1σ standard deviation) only show the upper bound for visibility. Significative difference (p-value <0.05) between the two seasons is given by the (\*) and the (†), respectively for the Welch two-sample test and the Wilcoxon sum rank test (see section 2.8 Statistical analysis for more details).**



**3.3 Seasonal differences**

The Shapiro-Wilk normality testing (Table S2) indicated that only a few environmental parameters were normally distributed in both the dry and wet seasons, when all the stations were grouped: temperature, DIC, $pH_T$, $[HCO_3^-]$, $[CO_3^{2-}]$, $\Omega_{calcite}$, $\Omega_{aragonite}$ and chlorophyll-*a*. Welch's two-sample t-tests conducted on these parameters (Table S3) revealed no statistically significant

differences (p-value > 0.05) between the two seasons, though some significant differences could be observed for individual stations: salinity at LA station and, $pH_T$, $pCO_2$, $\Omega_{calcite}$ and $\Omega_{aragonite}$ at station BLA. The Wilcoxon rank-sum test, applied to the remaining environmental variables and coccolithophore community data (Table S4), showed significant seasonal differences for only a few parameters, when all the stations were grouped: salinity, silicate, and precipitation. For the coccolithophore community, significant differences between the dry and wet seasons were found in total coccosphere abundance, the

concentrations of *E. huxleyi*, *G. oceanica*, several minor species, and the diversity index and richness. Similar differences were observed for the stations LA and BLA while the costal station LBW didn't show any significant difference between the two seasons.

**4 Discussion**

**4.1 Factors modulating the coccolithophore community off the Nigerian coast**

In the samples collected off the coasts of Lagos, the variations in total coccolithophore abundances (Fig. 2) and coccolithophore species richness and diversity (Table 2) show a clear seasonal pattern, with significantly higher abundance and diversity during the wet season (Table S4). The results of the PCA conducted on the whole dataset as well as for both the dry and wet seasons (Fig. 4) suggest a strong influence of precipitations on coccolithophore community structure. This is supported by the significant correlation between the total coccolithophore abundances (p-value < 0.01) as well as the abundances of *E. huxleyi*

(p-value < 0.05) and *G. oceanica* (p-value < 0.05) with precipitation (Table S5). However, no clear relationship was observed between environmental parameters and total abundances, diversity or richness when the whole dataset was considered (Table S5). Especially, no correlation with temperature (Dimiza et al., 2015; Giraudeau et al., 1993; Hagino et al., 2000) or nutrients (Balestra et al., 2017; Schiebel et al., 2004), which are known to influence coccolithophore distribution and diversity, was found across the entire dataset. This is further supported by the limited seasonal changes in most environmental parameters at

the sampling sites (Table S3 and S4), with most of the environmental variables showing little to no seasonal variations (Fig. 3).

Although not detected by the PCA, a mere 0.5°C seasonal temperature difference coincided with a marked shift in community composition, from *E. huxleyi* dominance in the wet season to *G. oceanica* dominance in the warmer dry season. The latter shows a significant correlation with temperature (Table S5). A significant negative correlation between assemblage diversity

and richness and temperature is also observed during the wet season (Table S5). This ecological shift aligns with culture studies where temperature was identified as a potential driver of species composition by differently affecting growth rates (Gafar and



Schulz, 2018). Notably, a global compilation of sediment coccolith abundances reported similar shifts around 27ºC, very close to the temperatures observed in our study (Gafar and Schulz, 2018).

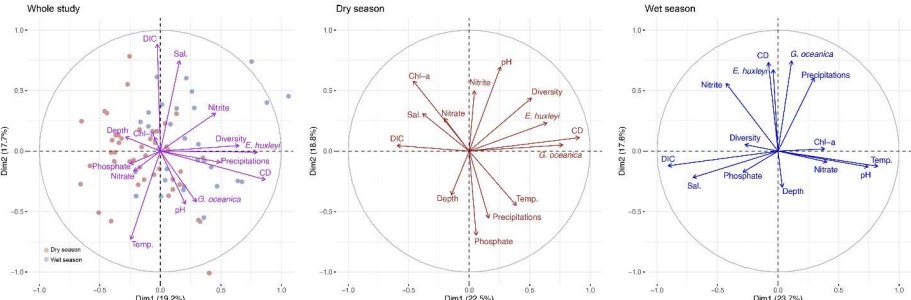

**Figure 4: An example for accurate data representation & universal readability of figures.**

**4.2 Phytoplankton dynamics off Nigerian coast**

Primary production in the ocean is sustained by phytoplankton functional types (i.e., diatoms, dinoflagellates and coccolithophores; O'Brien et al., 2013). Therefore, identifying the dominant coccolithophore taxa is essential. Chlorophyll-*a* concentrations, which in the open ocean typically reflect coccolithophore cell density (Balch et al., 2019), appear decoupled from coccolithophore abundances in this region (Fig. 4d–e; Table S5), suggesting that other phytoplankton groups must drive chlorophyll-*a* levels. Previous studies on phytoplankton communities in the inner shelf off Lagos have mainly focussed on diatoms (Nwankwo and Onyema, 2003), cyanobacteria (Onyema and Akanmu, 2018; Onyema and Popoola, 2013) and dinoflagellates (Akanmu and Onyema, 2020; Elegbeleye and Onyema, 2020), the latter making up only a minor contribution to the phytoplankton community. These studies established a diatom dominated seascape. Although maximum phytoplankton abundance during the dry season (i.e., diatoms, Onyema and Popoola, 2013) have been reported in the coastal waters off Lagos, our observations revealed lower coccolithophore concentrations during this period (Fig. 3e). We speculate that this may be linked to the general ecological succession between diatoms and coccolithophores in response to nutrient dynamics in coastal systems (Cermeño et al., 2011; Ziveri et al., 1995). Diatoms are known to be highly dependent on dissolved silica availability (Brzezinski et al., 2011; Xu et al., 2025). The pattern observed is supported by the higher average sea-water concentrations of silicate (Fig. 3f) during the dry season (7.42 ± 10.28 µmol L$^{-1}$) compared to the wet season (4.48 ± 3.33 µmol L$^{-1}$). This supports the hypothesis of a successional dynamics between these important phytoplankton functional groups. Although several coccolithophore species also require silicon, limiting silicate concentrations are an order of magnitude lower than the average silicate concentrations measured here (Langer et al., 2021). Therefore, it is likely that coccolithophores were not affected by silicate while diatoms were.



### 4.3 The role of the ITCZ

In our study site the coccolithophore standing stocks are mainly driven by the alternation of the wet and dry seasons (Table S5). As shown by the results of the Wilcoxon rank-sum tests (Table S4), the precipitation rate, nitrite concentration, the total abundances, diversity and richness of coccolithophore species have significant wet to dry season differences (p-value < 0.05). Precipitation pattern, indicative of the dry and wet seasons, was the only factor found to positively correlate with the total coccolithophore abundance during the entire study period (Table S5). The Gulf of Guinea is directly influenced by the oscillation of the ITCZ, which follows the position of the thermal equator (i.e., the latitude of the hottest air; Ali et al., 2011). During the wet season (over the study area), the ITCZ moves northward up to 15ºN, while during the dry season (over the study area), it moves southward down to approximately 5ºN (Fig. 5). In the area, the water masses general flow direction changes according to the position of the ITCZ, as revealed by the eastward seawater velocity (Global Ocean Ensemble Physics reanalysis, 2023) shown in Figure 5: during the dry season, the surface water masses tend to flow eastward, while during the wet season, they tend to flow westward. This suggests that the different coccolithophore communities observed during the dry and the wet season could have different geographical origins. During the dry season, the assemblage, dominated by *G. oceanica* could be brought offshore of Lagos by the intense Guinea current, possibly from the nearby upwelling system off Côte d'Ivoire and Ghana which shows a minor activity during the onset of the dry season (Ayissi et al., 2024). During the wet season, the assemblage dominated by *E. huxleyi* could be brought to the sampling site, from the nearby Niger Delta region, by an intensification of the westward current, possibly the Equatorial Undercurrent.

While the relatively short time series (~2.5 years) limits our ability to fully resolve ITCZ-driven interannual variability, our findings represent the first documentation of coccolithophore seasonality in this understudied region. These results are supported by robust statistical analyses and are consistent with the documented influence of the ITCZ on phytoplankton assemblages in nearby regions, such as the Cape Verde archipelago (Narciso et al., 2021; Silva et al., 2013) and the equatorial Atlantic (Guerreiro et al., 2019).



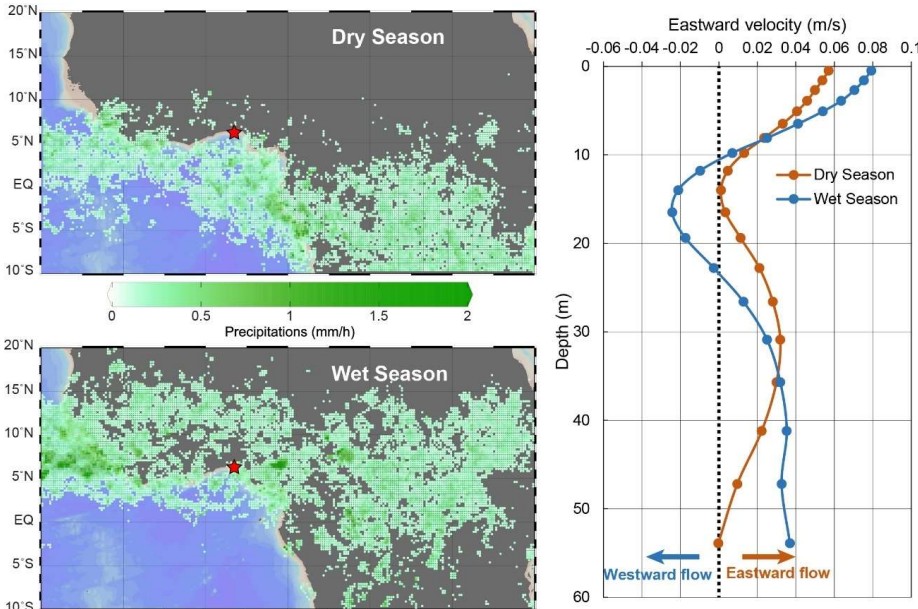

**Figure 5: ITCZ oscillation between the dry season (upper left) and the wet season (lower left) as depicted by the average precipitations (Huffman et al., 2019) for both seasons. Note that for visibility, average precipitations below 0.1mm/hr were not**
**mapped, the red star refers to the sampling location. Maps produced with Ocean Data View (Schlitzer 2023). On the right panel is shown the average eastward velocities (Global Ocean Ensemble Physics Reanalysis, 2023) of the water masses at the sampling location for the dry (brown) and wet (blue) seasons. Negative values show a westward flow while positive values show an eastward flow.**

### 4.4 The Gulf of Guinea and global coastal assemblages

Previous studies on coastal coccolithophore assemblages highlight significant variability driven by seasonal cycles, nutrient availability, and local oceanographic processes, all of which influence coccolithophore abundance and distribution (Addante et al., 2023; Godrijan et al., 2018). Although standing stocks can differ greatly depending on the specific coastal setting, our data indicate that the maximum abundances observed off Lagos are comparable to those reported in other shallow (<100m) coastal regions (de Vries et al., 2020; O'Brien, 2012). In these environments, coccolithophores coexist and compete with other

phytoplankton groups, which may limit their proliferation, leading to shifts in community composition and moderate cell division rates. Despite very different oceanographic and climatic settings represented in the global dataset we extracted (1780 samples, Fig. 6), coccolithophore standing stocks off Lagos (mean: $1.3 \times 10^4 \pm 1.0 \times 10^4$ cells L$^{-1}$; max: $4.8 \times 10^4$ cells L$^{-1}$ during the wet season) are of the same order of magnitude as those found in other coastal environments such as the South Pacific Ocean ($2.1 \times 10^4 \pm 5.7 \times 10^4$ cells L$^{-1}$), the Barents Sea ($3.0 \times 10^4 \pm 16.5 \times 10^4$ cells L$^{-1}$), or the Mediterranean Sea ($5.5$

$\times 10^4 \pm 7.5 \times 10^4$ cells L$^{-1}$) (Fig. 6; Table S8). These similarities suggest that coccolithophore production in coastal systems




may be subject to a set of common ecological constraints, such as higher turbidity, shallower mixed layers, greater nutrient fluctuations or even local pollution, regardless of the regional context.

In our study, both species richness and diversity fall in the lower range of those reported in comparable coastal settings (e.g., de Vries et al., 2020) and are markedly lower than values typically observed in oligotrophic oceanic regions such as the central
Mediterranean Sea (Ziveri et al., 2014). This low diversity may reflect environmental limitations typical of coastal tropical zones, including variable salinity, episodic nutrient inputs, and high competition with other phytoplankton groups such as diatoms and cyanobacteria. Additionally, seasonal hydrodynamics and changes in water mass origin, as discussed in section 4.3, may further limit the establishment of a more diverse coccolithophore community.

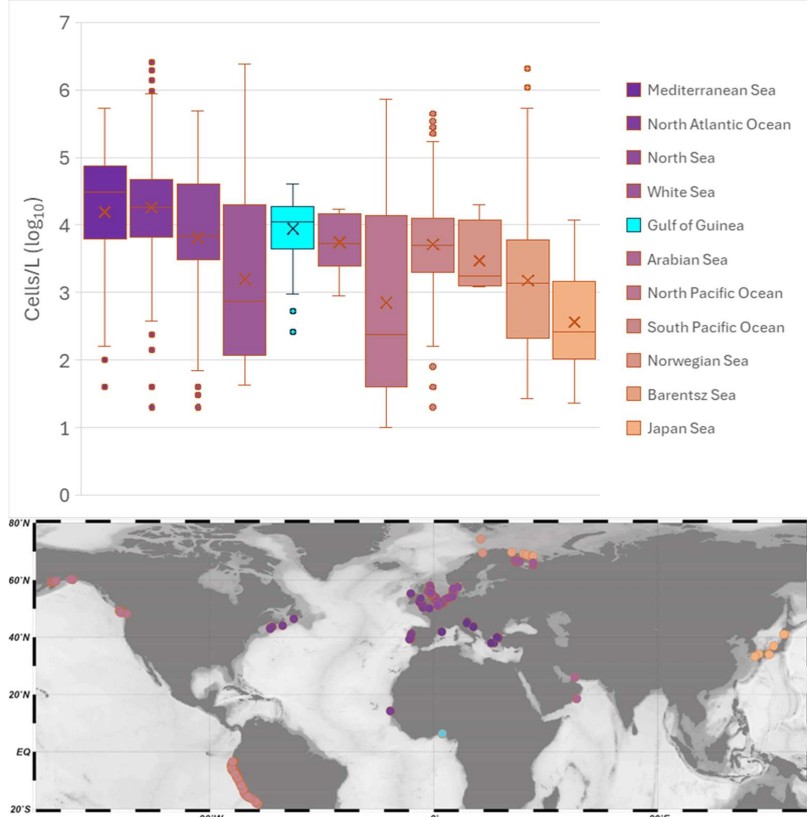

**Figure 6: Boxplots showing the distribution of cell concentrations (cells L⁻¹) for each main oceanographic basin. Each box represents the interquartile range (1st to 3rd quartile), with the horizontal line indicating the median and whiskers extending to the minimum and maximum values within 1.5× the interquartile range. Outliers are shown as dots. Crosses indicate the average of each series. The y-axis is on a logarithmic scale (log₁₀) to account for the high variability and wide range of concentrations across series. On the map are reported the locations of the samples used for each oceanographic basin.**



**5 Conclusions**

This study provides the first multi-seasonal assessment of coccolithophore abundance, diversity, and community structure in the coastal waters off Lagos, within the Gulf of Guinea. Our results reveal that coccolithophore standing stocks in this understudied tropical coastal region are comparable in magnitude to those observed in disparate coastal systems globally, despite notable differences in climatic and oceanographic settings. Seasonal variability driven by the Intertropical Convergence

Zone (ITCZ), reflected in changes in precipitation, nutrient concentrations, and surface currents, emerges as a key environmental driver of coccolithophore community composition. While the wet season is associated with higher coccolithophore abundances, particularly of *E. huxleyi*, the dry season supports distinct assemblages dominated by *G. oceanica*, likely influenced by shifts in water mass origin. Interestingly, chlorophyll-a concentrations, which are often used as proxies for coccolithophore biomass in the open ocean, appear decoupled from coccolithophore abundance in this coastal

setting, highlighting the influence of other phytoplankton groups and local ecological dynamics. Overall, our findings support the hypothesis that coccolithophore assemblages in tropical coastal systems are shaped by a shared set of ecological pressures that constrain both abundance and diversity. These constraints appear to operate independently of regional characteristics, suggesting common mechanisms linked to environmental variability, resource competition, and hydrodynamic conditions. Future research integrating long-term time series, finer taxonomic resolution, and broader geographic coverage will be critical

to better understand the ecological functioning of coccolithophores in coastal tropical environments and their role in regional biogeochemical cycles.

**6 Data availability**

All the original data used and presented in this manuscript are available in the supplementary tables.

**7 Author contribution**

**F.A.** participated in conceptualization, formal analyses, investigation, methodology and writing, reviewing and editing of the manuscript. **M.G.** participated in conceptualization, formal analyses, sample analyses, supervision, writing original draft, writing, review and editing manuscript. **G.L.** participated in conceptualization, methodology and assisted in writing, reviewing, and editing the original draft. **L.C.** assisted in conceptualization, investigation, supervision, writing original draft and writing and editing manuscript. **M.A.** carried out sample analyses, writing original draft, review and editing manuscript. **S.O.** assisted

in investigation, supervision, writing original draft, writing, reviewing and editing. K.S. contributed to manuscript writing, reviewing and editing. **P.Z.** assisted in conceptualization, methodology, assisted in funding acquisition, investigation, sample analyses, supervision, writing original draft, writing, review and editing manuscript.



**8 Competing interests**

The authors declare that they have no conflict of interest.

**9 Acknowledgements**

We appreciate the Nigerian Maritime Administration and Safety Agency (NIMASA) for the provision of MV Sea Marchall for field campaigns. The Nigerian Institute for Oceanography and Marine Research (NIOMR) is also thanked for providing access to laboratory for the preparation of seawater for coccolithophore samples.

**10 Financial support**

The authors acknowledge the financial support from the Ocean Foundation (TOF), ICTA-UAB "María de Maeztu" Programme for Units of Excellence funded by the Spanish Ministry of Sci-ence, Innovation and Universities (MDM-2015-055; CEX2019-000940-M), the Marine and Envi-ronmental Biogeosciences Research Group (MERS) Generalitat de Catalunya (2021 SGR 00640), and Spanish Ministry of Science and Innovation, BIOCAL Project (PID2020-113526RB-I00). GL acknowledges funding from the Spanish Ministry of Universities through a Maria Zambrano grant. The inorganic chemical oceanography

analysis were done by INOCEN (IEO-CSIC) lab technicians E.F. Guallart (PTA2016-12441-I), M. Castaño, J.G. Dequidt & R. A. Amigo (PEJ2018-003991-A).

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
