# Peer review of "Wet and dry seasons modulate coastal coccolithophore dynamics off South-western Nigeria (Gulf of Guinea)"

_EGUsphere, 2025_

## Author Response (AR1)

Dear Professor Koji Suzuki,

On behalf of my co-authors, I would like to thank you, Dr. Xiaobo Jin, and the anonymous reviewers for the constructive comments provided on our manuscript "Wet and dry seasons modulate coastal coccolithophore dynamics off South-western Nigeria (Gulf of Guinea)".

In the following, we provide a detailed, point-by-point response to all comments in the order they were published (i.e., CC1 (p. 2), RC1 (p. 4), and RC2 (p. 6)). The reviewers' comments are presented in italic, while our replies are given in plain text. All line numbers in our responses refer to the revised version of the manuscript without track changes.

We hope that our revisions and clarifications address the reviewers' concerns and improve the manuscript in line with their suggestions.

Sincerely,

Dr. Michaël Grelaud

(on behalf of all co-authors)

*CC1: 'Comment on egusphere-2025-3201', Xiaobo Jin, 13 Aug 2025:*

*The authors presented seasonal changes in living coccolithophore community off the coastal Nigeria. They found E. huxleyi dominated in wet seasons, and G. oceanica in dry seasons. This finding is interesting, with significant implications for coccolithophore ecology and biogeography.*

*Here I would present some comments for such alternating occurrences of the species. The authors found higher chlorophyll-a and silicate concentrations in the dry seasons, which may suggest a dominance of diatom in the phytoplankton community. Therefore, I speculate that the occurrence of E. huxleyi may result from their competition with diatoms. E. huxleyi cannot outcompete against diatoms in the dry seasons, and they would be found in the wet seasons. In contrast, G. oceanica may be a more efficient nutrient assimilator, and they would co-exist with diatoms. A similar case can be found in the South China Sea as reported in Jin et al. (2019, JGR-BG) and Jin et al. (2022, JGR-BG). The biogeography and seasonal production succession of E. huxleyi and G. oceanica can be governed by their competition with diatoms.*

**Reply on CC1:**

We thank Dr. Jin for the insightful comment regarding the alternating occurrence of *E. huxleyi* and *G. oceanica* in our study. We agree that there are similarities with the succession patterns revealed by sediment traps in the South China Sea (Jin et al., 2019, 2022). In both systems (considering stations TJ-G and TJ-E in Jin et al., 2019), *G. oceanica* co-occurs with diatoms (e.g., December in Jin et al., 2019, and the dry season in our study), followed by *E. huxleyi* dominance (e.g., late February in Jin et al., 2019, and the wet season in our study). We have included the suggested references in the revised manuscript (Section 4.2, "Phytoplankton dynamics off the Nigerian coast", line 305) when discussing the general ecological succession between diatoms and coccolithophores in coastal systems.

We also agree that silicate availability plays a key role in shaping diatom presence. In our study, silicate is the only nutrient that shows a significant seasonal difference (Fig. 3, Table S4), while all other nutrients remain relatively stable and quite high across seasons (Fig. 3, Tables S3–S4). Since *G. oceanica* has strong affinities with high-nutrient conditions (Andruleit et al., 2003; Andruleit & Rogalla, 2002), its occurrence should remain relatively consistent at our study site, in line with the year-round stability of most nutrients, which is not the case.

Moreover, there are notable differences in timing and amplitude between the two systems. In the South China Sea, the alternation occurs mainly during winter (December–March; Jin et al., 2019), reflecting a sequential bloom, with low coccolithophore production during the rest of the year. In contrast, in the Gulf of Guinea, coccolithophores are present year-round, with abundances peaking during the wet season (Fig. 2). This alternation follows the large-scale seasonal cycle driven by the latitudinal displacement of the ITCZ, which strongly influences the regional oceanography. All this together, along with the direct influence of the ITCZ on regional oceanography, supports our interpretation that the observed seasonal assemblages are not only the result of local nutrient – phytoplankton interactions, but they likely reflect broader hydrographic controls, with wet and dry season communities being shaped by distinct water mass influences.

References (added to the reference list, lines 525-530):

Jin, X. B., Zhao, Y. L., Zhang, Y. W., Wen, K., Lin, S., Li, J. R., Liu, Z. F. (2019). Two Production Stages of Coccolitho-phores in Winter as Revealed by Sediment Traps in the Northern South China Sea. JGR Biogeosciences, 124(7), 2335-2350. Doi: 10.1029/2019JG005070

Jin, X., Liu, C., Xu, J., Guo, X. (2022). Coccolithophore Abundance, Degree of Calcification, and Their Contribution to Particulate Inorganic Carbon in the South China Sea. JGR Biogeosciences, 127(4), e2021JG006657. Doi: 10.1029/2021JG006657

*The manuscript, "Wet and dry seasons modulate coastal coccolithophore dynamics off South-western Nigeria (Gulf of Guinea)" by Falilu Adekunbi et al., presents a valuable dataset acquired from three coastal stations between 2018 and 2021. The study's objective is to enhance our understanding of living coccolithophore dynamics, particularly in coastal environments, by examining their relationship with environmental variables and oceanographic settings in the Gulf of Guinea. The authors effectively use a statistical analysis approach to highlight and clarify the key relationships between taxa and environmental variables.*

*The data acquisition is described in detail, and the findings are well-illustrated with an adequate number of high-quality figures. The references are up to date.*

*All my comments and suggestions are intended to improve the quality of the paper by clarifying specific sections of the text. I recommend the manuscript for publication after minor revision.*

**Reply:**

We thank the Anonymous Referee #1 for their constructive remarks and positive feedback.

*Minor Suggestions*

*Hyphenation: Several instances of hyphenated words appear to be incorrectly divided across lines (e.g., line 29). The authors should carefully check the entire text to correct all such instances.*

**Reply:**

The issue of hyphenated words originates from text formatting and has been corrected in the revised version.

*Absence of Holococcolithophores: Lines 78-79 state, "Only heterococcolithophores were identified in all the samples (15 taxa, Table S1), as no holococcolithophore could be observed." While this is an interesting result, the authors do not provide an explanation for the absence of holococcoliths in the studied sites. Without this clarification, a reader (such as myself) might question the sampling strategy. For instance, the storage conditions mentioned in line 110 ("The seawater samples for coccolithophore community composition were stored shortly (<5 hours)") are critical and could be a contributing factor (e.g., exposure to sunlight or high temperatures). The authors should address this to rule out potential sampling issues and provide a more complete scientific explanation.*

**Reply:** Regarding the absence of holococcolithophores, we are confident that this is not related to sampling strategy. Standard protocols for coccolithophore analysis were followed, with samples processed within 5 hours of collection, and both light and scanning electron microscopy were used, minimizing the likelihood of missing holococcolithophores (liths or spheres) if they had been present.

We have clarified, in the revised manuscript (section 4.1 "Factors modulating the coccolithophore community off the Nigerian coast"), that their absence is most likely ecological by adding the following text and references (lines 284-287):

"Finally, the absence of holococcolithophores, the haploid phase of coccolithophores, can be explained by the fact that they are typically associated with stable, nutrient-poor open-ocean conditions. In contrast, the eutrophic and dynamic coastal waters off Lagos are not limiting for the heterococcolithophores, the diploid phase of coccolithophores, making

holococcolithophores unlikely to occur in our samples (Guerreiro et al., 2023; Penales et al., 2025)."

References (added to the refrence list, lines 510-512 and 587-590):

Guerreiro, C. V., Ferreira, A., Cros, L., Stuut, J. B., Baker, A., Tracana, A., Pinto, C., Veloso, V., Rees, A. P., Cachão, M. A. P., Nunes, T., & Brotas, V. (2023). Response of coccolithophore communities to oceanographic and at-mospheric processes across the North and Equatorial Atlantic. Frontiers in Marine Science, 10, 1119488. doi:10.3389/fmars.2023.1119488

Penales, P. J. F., Skampa, E., Dimiza, M. D., Parinos, C., Velaoras, D., Pavlidou, A., Oikonomou, V. A., & Triantaphy-llou, M. V. (2025). Coccolithophore Assemblage Dynamics and Emiliania huxleyi Morphological Patterns During Three Sampling Campaigns Between 2017 and 2019 in the South Aegean Sea (Greece, NE Mediterrane-an). Geosciences, 15(7), 268. doi:10.3390/geosciences15070268

**RC2: 'Comment on egusphere-2025-3201', Anonymous Referee #2**

*The manuscript "Wet and dry seasons modulate coastal coccolithophore dynamics off South-western Nigeria (Gulf of Guinea)" by Adekunbi et al. explores the coccolithophore dynamics at the Nigerian coast (Gulf of Guinea) during the contrasting environmental conditions related to dry and wet seasons. Overall, the manuscript is very well written, data analysis and presentation are of high quality, and the dataset (both coccolithophore abundances and environmental data) is of interest for the wider coccolithophore research community. I recommend the manuscript for publication in Biogeosciences and suggest some revisions that I believe would improve the quality of the manuscript:*

**Reply:**

We appreciate and thank the Anonymous Reviewer #2 for their very supportive comments and insightful feedback towards the improvement of our manuscript.

*General comment*

*- There are instances of incorrect hyphenation/line breaks that should be corrected, e.g. Lines 29, 391, 392, and several cases in the References.*

**Reply:**

The issue of hyphenated words originates from text formatting and has been corrected in the revised version.

*Introduction: A final paragraph stating the aims, hypotheses and impacts/relevance of the study is missing in the introduction section. I suggest the authors develop the final introduction paragraph from the sentence starting "In this study, for the first time…" (Line 63).*

**Reply:**

The end of the final paragraph of the introduction has been revised and extended to include the aims and relevance of the study as follow (Lines 64-67):

"In this study, we present the first periodic monitoring of coccolithophore communities and associated environmental parameters in coastal waters off Lagos, Nigeria, from December 2018 to April 2021, encompassing both the wet and dry seasons, characteristic of the region. The objectives were (1) to assess seasonal variability in coccolithophore abundance and community structure, and (2) to identify the environmental factors driving these dynamics, with the overarching goal of advancing our understanding of the ecology of this key group of marine primary producers in the Gulf of Guinea."

*Materials and methods*

*- Line 73: incorrect unit format "m.s.-1".*

**Reply:**

The format has been corrected in the revised manuscript (m.s$^{-1}$, line 75).

*- In the Study area subsection, the authors discuss the months that belong to wet or dry season and argue that the duration of the seasons has changed in recent years. They should also state in the Methods which system was used to define wet or dry seasons in this study. This is shown in Table S6, but it should also be clarified in the Methods section. Was the Wet/Dry distinction*

*predefined, or was it based on actual precipitation data from the studied period? If the latter is the case, it should be further clarified in the Methods section.*

**Reply:**

We agree that a clarification is needed here. Indeed, the Wet/Dry season distinction was predefined in order to combine the "historical" seasonality (Nwankwo et al., 1996; Dry: November – April; Wet: May – October) and the fact that this seasonality has been shifted with the wet season that can extend up to November (Fasona et al., 2019). This has been clarified in the revised manuscript (last paragraph of the section 2.1 Study area, lines 86-93) as follow:

"The rainfall in the Lagos coastal environment is characterized by a double maximum pattern where according to Nwankwo (1996) the dry season commences in November and extends until April while the wet season is from May to October with a break in August (Fig. 1). However, climate change has resulted already in local shifts in rainfall pattern, intensity and frequency, consequently the wet season now extends to November reaching a peak in July and September (Fasona et al., 2019). To take into account this shift, in this study the dry season is defined from December to April while the wet season is defined from May to November."

- Line 103: Change the salinity unit to PSU, also in Fig. 3B.

**Reply:**

The salinity unit has been changed to PSU in both main text (line 106) and figure (see below) in the revised manuscript.

[Figure]

Revised Figure 3.

*- Samples for SEM analysis were collected, but no SEM images or SEM data are shown. Were the 15 species identified only under Polarized LM analysis, or were they also observed by SEM? If there is SEM data available confirming the taxonomic identification of coccolithophore species found under LM analysis, consider adding these images to the supplement and listing any additional species observed under SEM. If SEM data is not included in the manuscript, perhaps you can remove the mention of sampling for SEM and SEM sample preparation from the Methods section altogether.*

**Reply:**

SEM was indeed used to confirm the species identified through light microscopy. SEM pictures of all the identified species have been added to a new section of the supplementary material word file as Figure S1. Moreover, reference to Fig. S1 has been added in the revised manuscript in section "3.1 Coccolithophore distribution" (line 179). During the process, we managed to identify the minor species "Syracosphaera sp." as *Syracosphaera prolongata*. The species' name has been replaced in tables S1, S2, S3, S4 and S6.

[Figure]

Figure S1: Scanning electron Microscope (SEM) photomicrographs of coccolithophore off Lagos coastal waters A) *Emiliania huxleyi*, B) *Gephyrocapsa oceanica*, C) *Gephyrocapsa ericsonii*, D) *Syracosphaera tumularis*, E) *Calciopappus rigidus*, F) *Umbilicosphaera hulburtiana*, G) *Discosphaera tubifera* (coccolith), H) *Michaelsarsia*

*elegans*, I) *Calsiosolenia corselii*, J) *Helicosphaera carteri*, K) *Syracosphaera prolongata*, L) *Florisphaera profunda*, M) *Syracosphaera histrica*, N) *Umbellosphaera tenuis*.

*- Line 126: word "young" should be removed?*

**Reply:**

The word "young" has been removed in the revised manuscript (line 129).

*- Line 129: "2.4. Carbonate chemistry variables" should be removed*

**Reply:**

It has been removed in the revised manuscript (line 132).

*- Line 164: "2.8. Statistical analyses" should be removed*

**Reply:**

It has been removed in the revised manuscript (line 167).

- Line 165-166: unclear sentence without a full stop.

**Reply:**

It has been removed in the revised manuscript (line 167).

*Results*

*- Consider adding a supplementary figure showing LM images of 15 species observed during the analysis of the coccolithophore community if the images are available.*

**Reply:**

Unfortunately, LM images of coccolithophores are not available, however, SEM images have been provided as Figure S1 of the supplementary material.

*- Data from Table 2 would be easier to read if presented as a Figure/plot. Abbreviations for station labels should be provided in the caption rather than referring to the Table 1.*

**Reply:**

The data of Table 2 are now visually presented in Figure S2 (see below) of the supplementary material. Reference to Fig. S2 has been made in the revised manuscript in the last paragraph of the section 3.1 Coccolithophore distribution (line 211). Moreover, the abbreviations have been clarified in the caption of Table 2 (i.e., BLA: Beyond Lagos Anchorage, LA: Lagos Anchorage, LBW: Lagos Break-Water; lines 216-217).

[Figure]

**Figure S2: Average seasonal diversity index (top) and species richness (bottom) for three stations (BLA: Beyond Lagos Anchorage, LA: Lagos Anchorage, LBW: Lagos Break-Water) and all stations together (All).**

*- Figure 3 caption: should begin with "Environmental parameters…"*

**Reply:**

"An" has been deleted in the revised manuscript (line 240).

*- It is unclear whether the subsection "3.3. Seasonal differences" provides new data or summarizes environmental data from the previous section. If the latter is the case, I suggest you incorporate seasonal differences in coccolithophore community composition and environmental variables into previous subsections that are focusing on the respective topics.*

**Reply:**

The section "3.3 Seasonal differences" presents the results of the statistical analysis that was carried out on our dataset. To make it clearer, we have changed the title of the section for "3.3 Statistical assessment" in the revised manuscript (line 248). However, we prefer to keep this section separated from sections 3.1 and 3.2 which are more descriptive and separated between the coccolithophores and the environmental parameters, while the section 3.3 highlights the significant seasonal differences for all the parameters (biotic and abiotic) providing the base for our interpretations in the discussion.

Moreover, following another comment raised by the reviewer about the 1st section of the discussion ("4.1. Factors modulating the coccolithophore community off the Nigerian coast"; see below), we briefly introduce, in section 3.3, the results of the Pearson correlations performed between the environmental parameters and the total abundances of *E. huxleyi*, *G. oceanica*, coccolithophores, Shannon index and richness (Table S5) as well as the Principal Component Analysis (PCA) (lines 257-262):

"[…] Similar differences were observed for the stations LA and BLA while the coastal station LBW did not show any significant difference between the two seasons. Finally, the results of the Pearson correlations (Table S5) suggest that precipitation is the only environmental parameter showing significant correlation with the total coccolithophore abundances, and to a lesser degree with the abundances of E. huxleyi and *G. oceanica*. Interestingly, the highest correlation is observed between the abundances of *G. oceanica* and sea temperature during the wet season, which is not highlighted by the results of the PCA (Fig. 4)."

*- Figure 4: Caption is inadequate. Also, consider rearranging the three PCA plots the same way as in Fig. 2 (underneath each other) and making them larger and more readable. I also suggest you use different color coding for species names and environmental drivers in the PCA figures (if technically possible).*

**Reply:**

We apologize for this mistake regarding the caption of Figure 4. The caption should read (lines 290-292):

"Figure 4: Results of the principal component analysis (PCA) between the selected environmental variables, selected coccolithophore species and the diversity index, performed for the whole study (purple), the dry season (brown) and the wet season (blue). Sample distributions are shown for the whole study for both the dry (red) and wet (blue) seasons."

Figure 4 has been reorganized vertically (see below) as suggested by the reviewer. However, we could not change the colour for the species names.

[Figure]

Revised Figure 4

*- Figure 6. Typo – Barents Sea is written wrong ("Barentz")*

**Reply:**

The Figure 6 has been corrected (see below).

[Figure]

Revised Figure 6.

*Discussion*

*- Some parts of the first subsection of the discussion ("4.1. Factors modulating…") read more like the results section, providing novel results and referring to tables that were not referred to or reported in the Results section. Consider revising these parts in order to have better distinction between Results and Discussion.*

**Reply:**

All the tables mentioned in the section 4.1 were introduced in section 3.3 Statistical assessment (former section "3.3 Seasonal differences"), except for Table S5. This has been addressed in a previous comment (lines 257-262). Now the results of the tables S3, S4 and S5 are presented in the revised section 3.3 as well as some results of the PCA (Fig. 4).

*- Line 339: Consider that massive blooms of Emiliania huxleyi are regularly reported in the Barents Sea (e.g. Giraudeau et al 2016,). Are any large scale coccolithophore blooms reported*

*from the Gulf of Guinea? The standing stocks may be generally comparable, but the two systems are arguably very different when it comes to coccolithophore dynamics.*

**Reply:**

To date, no coccolithophore bloom has been reported in the Gulf of Guinea. We acknowledge that the systems compared in this section are very different from one to another. This is what we were suggesting by saying: "Despite very different oceanographic and climatic settings represented in the global dataset we extracted […], coccolithophore standing stocks off Lagos […] are of the same order of magnitude as those found in other coastal environments such as the South Pacific Ocean […], the Barents Sea […], or the Mediterranean Sea […]."